# Potential Access to Emergency General Surgical Care in Ontario

**DOI:** 10.3390/ijerph192113730

**Published:** 2022-10-22

**Authors:** Jordan Nantais, Kristian Larsen, Graham Skelhorne-Gross, Andrew Beckett, Brodie Nolan, David Gomez

**Affiliations:** 1Li Ka Shing Knowledge Institute, St. Michael’s Hospital, Toronto, ON M5B 1T8, Canada; 2Department of Surgery, Section of General Surgery, University of Manitoba, Winnipeg, MB R3A 1R9, Canada; 3CAREX Canada, Faculty of Health Sciences, Simon Fraser University, Vancouver, BC V6T 1Z3, Canada; 4Department of Geography and Planning, University of Toronto, Toronto, ON M5S 3G3, Canada; 5Department of Geography and Environmental Studies, Ryerson University, Toronto, ON M5B 2K3, Canada; 6Department of Surgery, Division of General Surgery, University of Toronto, Toronto, ON M5T 1P5, Canada; 7Department of Medicine, Division of Emergency Medicine, University of Toronto, Toronto, ON M5S 3H2, Canada; 8Institute of Health Policy, Management & Evaluation, University of Toronto, Toronto, ON M5T 3M6, Canada

**Keywords:** emergency general surgery, potential access, access to care, health care models, geographic information system, rural surgery

## Abstract

Limited access to timely emergency general surgery (EGS) care is a probable driver of increased mortality and morbidity. Our objective was to estimate the portion of the Ontario population with potential access to 24/7 EGS care. Geographic information system-based network-analysis was used to model 15-, 30-, 45-, 60-, and 90-min land transport catchment areas for hospitals providing EGS care, 24/7 emergency department (ED) access, and/or 24/7 operating room (OR) access. The capabilities of hospitals to provide each service were derived from a prior survey. Population counts were based on 2016 census blocks, and the 2019 road network for Ontario was used to determine speed limits and driving restrictions. Ninety-six percent of the Ontario population (*n* = 12,933,892) lived within 30-min’s driving time to a hospital that provides any EGS care. The availability of 24/7 EDs was somewhat more limited, with 95% (*n* = 12,821,747) having potential access at 30-min. Potential access to all factors, including 24/7 ORs, was only possible for 93% (*n* = 12,471,908) of people at 30-min. Populations with potential access were tightly clustered around metropolitan centers. Supplementation of 24/7 OR capabilities, particularly in centers with existing 24/7 ED infrastructure, is most likely to improve access without the need for new hospitals.

## 1. Introduction

Emergency general surgery (EGS) describes a broad grouping of infectious, obstructive, and hemorrhagic pathologies of the abdomen treated by the general surgeon on an emergent basis including both operative and non-operative management [1,2]. EGS diseases are responsible for 7.1% of US hospital admissions, and these patients have more than 7 times the mortality of elective general surgical patients [3]. Delays to treatment are one potential contributor to worsened mortality and complications in EGS conditions [4,5,6,7]. Additionally, access to 24-h EGS care and operative rooms is associated with improved outcomes for some life-threatening EGS conditions such as bowel perforation and necrotizing infection [8].

EGS care is not consistently regionalized, and patients typically receive treatment at their closest hospital, with only a small proportion being transferred [9]. Furthermore, 87% of patients self-transport to the emergency department [10], presumably driving or using public transport indicating that road travel time is the most important determinant of potential access in EGS conditions.

The use of geographic information systems (GIS) analysis, which applies computerized tools to understand clinical data in a geospatial (typically map-based) context [11], is a well-established and validated method for estimating road travel times [12,13]. Given the relationship between access to EGS care and outcomes, understanding the distribution and capabilities of centers providing this care, relative to the population, is a necessary first step in improving the associated system. The purpose of this study was to address this knowledge gap by using GIS analysis to estimate the potential access of people in Ontario to EGS care, including around-the-clock emergency and operating room access.

## 2. Principles of Healthcare Access and Organization

### 2.1. Access to Healthcare

Access to healthcare is a multi-faceted concept with varied definitions, and typically includes a complex interplay between the characteristics of the individual accessing care and the healthcare system [14,15,16,17]. A frequently accepted definition was conceptualized by Levesque et al., (2013) and breaks access down into five dimensions necessary to generate access: approachability, acceptability, availability and accommodation, affordability, and appropriateness [14].

Each of these dimensions reflects a potential barrier to access for individuals. Approachability refers to the ability of people in need of a service to identify that the service exists and could benefit them. Acceptability is a measure of how the potential service interacts with the social and cultural norms of patients. Availability and accommodation relate to the ability of patients to physically reach needed services in a timely manner. The economic constraints affecting healthcare access fall into the category of affordability. Finally, appropriateness deals with the suitability of the described services in addressing the specific health problem of a particular individual [14].

Gaining a comprehensive understanding of access to healthcare requires that each of these dimensions are studied and addressed [14]. Resultantly, each individual component of this framework is necessary but not sufficient to quantify access to healthcare for individuals. The true measurement of access therefore requires in-depth study and quantification of each dimension. Despite this, evaluation of the ability of individuals to physically reach treatment is a necessary first step to understanding healthcare access, as no resolution of the other dimensions of care will result in adequate treatment without this component.

### 2.2. Accessibility of Emergency Medical Services

An important subset of healthcare access is the ability of patients to receive emergency medical services [18]. Due to the time-critical nature of emergency medicine, there is an observed relationship between delays in accessing appropriate healthcare and associated interventions and worsened outcomes including mortality [19]. This is particularly evident in disease-processes which can rapidly progress to death or significant morbidity, such as myocardial infarction, stroke, or trauma [20,21,22,23].

Amongst the determinants of healthcare access, proximity to an institution capable of managing a particular emergency condition has been shown to be directly related to outcomes [19]. This corresponds to the principle of availability and accommodation. Although it is difficult to ascertain the true ability of each individual to access emergency healthcare due to complexities in both the individual and systemic aspects of care, travel time can provide a useful analog measure to estimate this facet of access [13].

### 2.3. Emergency Medical Services in Ontario

Ontario is the most populous province in Canada with over 13 million inhabitants at the time of the 2016 census [24]. The province is over 900,000 km^2^, or 1/10th the size of the United States [24,25], and the population is predominantly urban, with less than 1/5th living in rural settings [24].

Emergency departments in Ontario experience nearly 6 million patient visits per year of varying severity [26]. These visits take place over an extensive land area with associated disparities in population density. Resultantly, emergency facilities range substantially in size and capabilities, from small rural centers to large tertiary academic institutions. Related to the remote location or small size of some of these centers, they may be able to offer some emergency services without having the ability to provide operative care. Hospitals in Ontario with operating and emergency room capabilities typically perform some number of urgent or emergent cases, including in general surgery. There are few private for-profit surgical suites and these are not typically associated with emergency facilities. Furthermore, operating rooms at smaller, non-tertiary centers may not have specialized services required for more complex cases. For patients with a known or suspected surgical illness, or requiring highly specialized procedures, this leads to an additional challenge to care acquisition as it delays many aspects of treatment until the patient reaches a capable center.

Ensuring that patients attain appropriate care for an acute illness is challenging in this setting, and may require that emergency transport bypasses some centers or transfers individuals between institutions. The complexities of this system highlight the potential difficulties in ensuring that patients receive timely intervention for life-threatening illnesses, and the numerous barriers to healthcare access involved.

## 3. Materials and Methods

### 3.1. Study Design

We performed a population-level GIS analysis of potential access to EGS care in the province of Ontario, Canada. We utilized a combination of census, survey, and road-network data to provide as complete of an analysis as possible.

### 3.2. Classification of Care Offered at Each Hospital

Hospital information was derived from a survey assessing structures, processes, and models of care across hospitals in Ontario in 2020 [27]. The authors identified that 114 of the 147 hospitals within Ontario offered urgent general surgical care: non-elective general surgery within 24–48 h of presentation, which we define here as EGS care [28]. The survey had a response rate of 96% (109/114) and provided information on the capabilities of individual institutions to offer various services. All 114 institutions were considered capable of providing EGS care based on publicly available information for the 5 non-respondents. We used survey data to further classify these institutions by whether they offered 24-h, any day (24/7) emergency department (ED) care (*n* = 100), or 24/7 operating room (OR) availability (*n* = 77), corresponding to critical aspects of EGS services [27]. Although some centers with 24/7 EDs did not have 24/7 ORs, all those with a 24/7 OR had around-the-clock EDs, making OR availability the most restrictive factor. In cases where survey data were not available, hospitals were assumed to not be capable of offering 24/7 ED or OR services. The capability to provide EGS care with 24/7 access to both ED and OR services was considered complete EGS access, as all of these factors are required for the provision of EGS treatment at any time.

### 3.3. Measuring Potential Access

The number of people with potential access was calculated based on the population reported at the 2016 census block level, the smallest geographic unit on which the Census of Canada releases information regarding the population [24]. The census block polygon file was converted to points based on the block centroid, for example a point in the center of the block represents the entire block population. Census blocks are very small units in urban and suburban areas but can be quite large in more rural environments. While there are limitations with this approach, it has been applied in similar studies to represent the population living within each block [12]. A number of census blocks makes up each census subdivision (CSD) which typically corresponds to a single municipality or community and is a convenient construct for the purposes of population description.

Potential access was measured using a GIS-based network analysis, examining travel times along the road network from the closest hospital (mapped to its exact address) to each block centroid. We created land travel catchment areas based on 15-, 30-, 45-, 60- and 90-min travel times and non-overlapping polygons (i.e., an individual can only be in one catchment area) [29]. This range of times was selected as 30-, 45-, and 60-min were felt to correspond to transport times allowing timely intervention in critical diseases and the additional times provide further context for interpretation of the results [12]. A network dataset for Ontario was computed using the 2019 road network which included data on speed limits and driving restrictions [30]. Travel time was used as the impedance for all catchment areas, meaning the fastest route would always be taken. These times accounted for speed limits and any driving restrictions.

All block centroids within the land travel catchment area were classified as having access to a center providing each type of care. Since non-overlapping travel catchments were created, population centroids would only be matched to one catchment in each instance (i.e., closest facility for that type of care). To calculate the proportion of each CSD with potential access at a set time, the sum of people in census blocks within that CSD able to potentially access an appropriate center within the denoted time was divided by the total population within the CSD. These steps were completed individually for each time examined.

Remote, fly-in only communities were identified and classified as not having road access to any type of care. Because some remote communities in Ontario can only be reached by typical means at certain times of the year (i.e., winter roads, ferries) [31], we assumed a “best-case scenario” wherein all calculations of potential access were performed when these routes were open.

We used the remoteness index (RI) to quantify how remote the population was within each CSD [32]. The RI was developed by Statistics Canada to quantify the remoteness of communities based on their size and proximity to major population centers and correlates with preventable mortality in populations with a low proportion of Indigenous inhabitants [32]. The value of the RI ranges from 0 to 1, with 1 being the most remote possible location [32].

### 3.4. Statistical Analysis

Potential access at each time interval was reported as both the number of persons and percentage of the overall Ontario population. Where land areas were reported, they were given as km^2^ and the percentage of the total land area. Continuous variables were non-normally distributed and therefore given as medians with interquartile ranges (IQR). Statistical tests were carried out using SAS v.9.4 (SAS Institute Incorporated, Cary, NC, USA) and Microsoft Excel Version 2203 Build 16.0.15028.20152 (Microsoft Corporation, Redmond, WA, USA).

## 4. Results

We examined 575 CSDs encompassing the entire population of Ontario. The CSDs had a median population of 2524 persons (IQR 461–10,162), ranging from no permanent population to 2,730,283 people in the most populous subdivision. The population density of individual CSDs varied notably, with the sparsest permanently populated area possessing a density of 0.014 persons per km^2^, the most densely populated CSD occupied by over 4300 persons per km^2^, and a median of 15 (IQR 4–62) persons per km^2^.

The median RI was 0.26 (IQR 0.12–0.42), indicating a relatively low degree of remoteness for the majority of the populace. We identified 17 fly-in only communities. Ten of these areas had some permanent population and held a total of 3986 people (population density 10 persons per km^2^), representing less than 0.1% of the Ontario population.

Hospitals offering any EGS care can potentially be accessed by 96% (*n* = 12,933,892) of the Ontario population within 30-min, increasing to 98% (*n* = 13,192,206) by 60-min (Table 1).

The populace with potential access to a 24/7 ED in 30-min was 95% (*n* = 12,821,747). This increased to 98% (*n* = 13,175,619) of the population at 60-min. Potential access to all EGS factors, including a 24/7 operating room, was more limited for a large portion of the population, as only 93% (*n* = 12,471,908) had potential access within 30-min, and 98% (*n* = 13,149,279) at 60-min. This corresponds to 349,839 people with potential access to a 24/7 ED without 24/7 OR availability at 30-min.

Potential access to any EGS care (A) and 24/7 OR (B) at 30-min for each CSD is displayed in Figure 1.

The highest levels of potential access to EGS care are centered around major metropolitan areas (Figure 1, named city centers), and the lowest levels of potential access correspond to remote regions. Although 95% (*n* = 12,820,808) of the population live in CSDs where >80% have potential access to EGS care within 30-min, these CSDs represent only 10% (Area = 94,160 km^2^) of the provincial landmass (Appendix A). Similarly, 90% (*n* = 12,050,424) of the population live in areas wherein >80% of the population has potential access to all factors, including a 24/7 OR, at 30-min, situated on a small percentage (8%, area = 77,233 km^2^) of land (Appendix A).

## 5. Discussion

We used a GIS network-based analysis to examine the potential access of the populace of Ontario to any EGS care, and two additional core components of related treatment: 24/7 ED and 24/7 OR availability. Our model showed that most of the population could potentially access a hospital with EGS capabilities within an appropriate period of time, but that reaching a hospital with 24/7 ED or 24/7 OR capabilities was more challenging for those living outside of major metropolitan areas. Since all centers with 24/7 ORs also had 24/7 EDs, OR availability appeared to be the rate limiting factor in many centers for potential access to 24/7 EGS care. Two percent of the population (*n* = 349,839) had potential access at 30-min to an ED to diagnose an EGS condition, without a 24/7 OR in which to treat it (Table 1). Although the gap narrowed with increasing timeframes, 28,420 people (<1%) remained with 24/7 ED but not 24/7 OR potential access even at 90-min. Potential access varied remarkably by location throughout the province (Figure 1) and was most challenging in rural or remote areas. Many of the most remote communities rely on provincial air ambulance services to provide transport to an EGS capable center; however, commonly there are delays associated with both the decision to transfer and the transfer itself (such as weather or no available asset) [33,34], making this a non-ideal strategy when suitable treatment can be offered locally.

Overall, these data were reassuring in that they indicate that EGS treatment was potentially accessible for the majority of the population. The finding that despite having potential access to an EGS center at 30-min, 1% (*n* = 112,145) cannot utilize an ED and 3% (*n* = 461,984) cannot utilize an OR at all hours is concerning. Operative intervention in EGS cases does not necessarily adhere to the predictable timing of more elective care. Nearly half of all operative EGS cases take place after hours and approximately 15% occur at night, reflecting the unpredictable nature of these diseases [4]. Since delays in operative management contribute to worsened morbidity and mortality in EGS, it stands to reason that centers offering EGS care without 24/7 OR access may be less able to provide timely and effective care. This suggests that some hospitals would benefit greatly from an expansion of after-hours capabilities. In particular, increasing OR hours in centers with established 24/7 EDs may be useful in mitigating delays in EGS treatment and those of other surgical diseases.

There were several limitations to our work. Census, survey, and road-network data were collected from differing but similar timepoints. The most recent data available at the time of collection was used for each category to minimize the effects of fluctuations over time. Despite this, it is expected that some degree of change may have occurred in the intervening time in each of these categories, which may limit the validity of the associated findings. Driving times represented an idealized scenario based on available road information. We could not account for disruptions to traffic flow, such as by inclement weather, and only road travel was examined. Furthermore, population with access was measured at the census block centroid, which may over or underestimate access in some areas. Services offered by Ornge, the sole provincial critical care transport and air ambulance service in Ontario [27], were not included. Air ambulances are critical in reaching rural and remote patients, and this may have unaccounted consequences for this group. However, this effect is expected to be small as EGS conditions only represent 9% of Ornge interfacility transfers [33].

## 6. Conclusions

Nearly all people had potential access to EGS care in an appropriate period of time by road. Fewer were able to access a center offering 24/7 operative and emergency capabilities. The most limiting factor appears to be 24/7 OR access, and an expansion of these abilities would likely be beneficial, particularly outside of metropolitan areas. The addition of operative capabilities to all centers providing emergency services would provide comprehensive EGS care to an additional 2% of the population of Ontario within 30-min, with nearly 350,000 people gaining improved access. This would require substantial investment in the 23 existing centers with disparate ED and OR access, but would provide a potentially profound improvement in the speed at which patients in rural areas could receive life-saving care.

## Figures and Tables

**Figure 1 ijerph-19-13730-f001:**
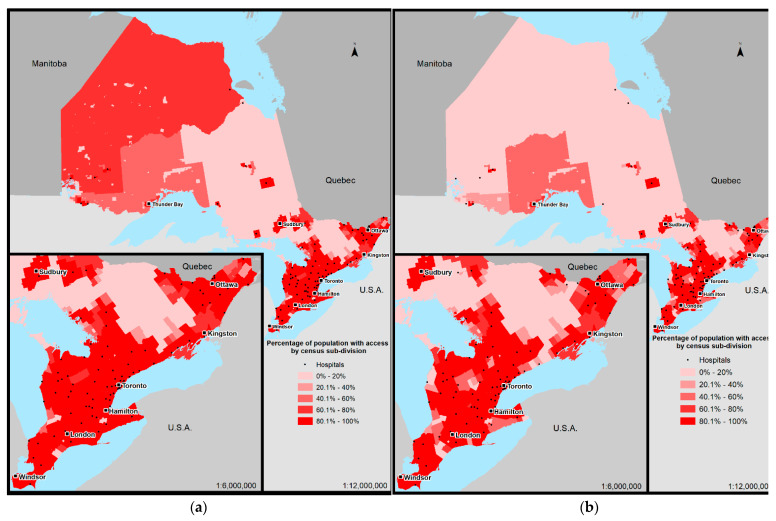
Percentage of Ontario population with access to any emergency general surgery care by census sub-division (**a**), versus those with access to all factors, including 24/7 operating room (**b**), within 30-min driving time. • denotes the location of a hospital, names denote major regional city centers.

**Table 1 ijerph-19-13730-t001:** Overall population with potential access to each type of care at varied driving time intervals.

		Population, *n* (% *) with Potential Access		
		Any Emergency General Surgery	24/7 Emergency Department	24/7 Operating Room (All Factors)
Time to Access (min)	15	11,527,813 (86)	10,734,954 (80)	9,779,865 (73)
	30	12,933,892 (96)	12,821,747 (95)	12,471,908 (93)
	45	13,131,755 (98)	13,115,851 (98)	13,077,515 (97)
	60	13,193,206 (98)	13,175,619 (98)	13,149,279 (98)
	90	13,251,229 (99)	13,245,585 (99)	13,217,165 (98)

* Percentage of total population of Ontario (2016 census), 13,436,631.

## Data Availability

Restrictions apply to the availability of these data. Data were obtained from DMTI Spatial and are available for purchase from DMTI Spatial.

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
