# Peer review of "Potential Access to Emergency General Surgical Care in Ontario"

_ijerph, 2022, doi:10.3390/ijerph192113730_

Round 1
Reviewer 1 Report
The article is an original study, which, after some corrections, could be of interest to many readers, because it deals with current and important from a scientific and practical point of view issues of the accessibility of medical services.
However, the article lacks the theoretical part, and the conclusions seem to be extremely limited and to a large extent repeat the results of the research.
That is why, I believe that the article would be much more valuable if the Authors considered supplementing the text with the following elements:
1) I suggest adding a separate theoretical part (where the Authors can explain the concept of access to health care and its dimensions) and an exhaustive literature review of research on the accessibility of emergency medical services;
2) in this part it is also worth presenting the organizational assumptions of the emergency medical system in Ontario, Canada (for an international reader it may be unclear when there is an emergency department in a hospital and there is no 24-hour operating room);
3) in this context, it should also be clarified if all ORs operate within the emergency medical system, or can they be, for example, orthopaedic or gynaecological operating rooms? (it is worth explaining the role and functioning of EDs and ORs in Ontario, Canada);
4) I suggest expanding conclusions with the consequences of the results obtained (also taking into account spatial aspects) and recommendations for the future (indication of places with insufficient level of access to EGS and recommendations where the units of the emergency medical system should be launched; how will launching of new units improve the level of accessibility?);
5) I recommend adding a map of catchment areas to the graphical presentation of the results – after all, it is the main result of the research procedure;
5) I did not find tables S1 and S2 to which the Authors refer in the text (page 5).
Despite the above-mentioned imperfections, I believe that the reviewed article is an interesting study that required from the Authors a considerable amount of work and a detailed diagnosis of the studied phenomenon. Once the suggested changes are introduced (or if doubts are otherwise cleared), it can become a valuable resource for researchers and health policy makers.
Reviewer 2 Report
The work is fine and it is always very positive to combine knowledge of health with geography. Anyway I notice several weaknesses.
On the one hand, the data are not very current because they speak of a 2016 Census. So, at most they will be from 2015, 7 years passed and during that time many events such as a pandemic such as COVID-19 that, with total certainty, modified the population values of Ontario. The 2019 wagon network data is also old.
With the added problem that the hospital data is from 2020. So there is a chronological heterogeneity that greatly weakens the work. This problem should be solved.
As a GIS specialist, they have to explain much more the weaknesses of using the centroids of the blocks. Well, a lot of information becomes homogeneous that is not. Within the studies of health geography, these generalities gain greater value, because we are representing people. Another type of mapping and analysis in detail should be considered. GIS allows it.
In addition, the work should be complemented by further study. Although it is fine, I see it a little superficial: they could represent the densities, both of population, as the capacity of each of the communication routes, values of traffic intensity, differentiation between the different hours of the day, etc.
So, in my view, if they explain the mismatch between the dates of the sources of information and even if they solve it. As well as the incorporation of some more variable in its analysis, an indicator on the population of Ontario, its infrastructure network or urban network, represent the points with traffic problems, etc. The work could be published.
Round 2
Reviewer 1 Report
The Authors considered most of my comments. The changes introduced to the text along with the Authors' explanations make me think that the paper should be published. I do not have any more remarks.